# Morphology of Anterior Cingulate Cortex and Its Relation to Schizophrenia

**DOI:** 10.3390/jcm12010033

**Published:** 2022-12-21

**Authors:** Anastasiya Lahutsina, Filip Spaniel, Jana Mrzilkova, Alexandra Morozova, Marek Brabec, Vladimir Musil, Petr Zach

**Affiliations:** 1Department of Anatomy, Third Faculty of Medicine, Charles University, 100 00 Prague, Czech Republic; 2National Institute of Mental Health, 250 67 Klecany, Czech Republic; 3Department of Statistical Modeling, Institute of Computer Science, Academy of Sciences of the Czech Republic, 182 00 Prague, Czech Republic; 4Centre of Scientific Information, Third Faculty of Medicine, Charles University, 100 00 Prague, Czech Republic

**Keywords:** MRI, schizophrenia, morphology, anatomy, cingulate and paracingulate sulci

## Abstract

Cortical folding of the anterior cingulate cortex (ACC), particularly the cingulate (CS) and the paracingulate (PCS) sulci, represents a neurodevelopmental marker. Deviations in in utero development in schizophrenia can be traced using CS and PCS morphometry. In the present study, we measured the length of CS, PCS, and their segments on T1 MRI scans in 93 patients with first- episode schizophrenia and 42 healthy controls. Besides the length, the frequency and the left-right asymmetry of CS/PCS were compared in patients and controls. Distribution of the CS and PCS morphotypes in patients was different from controls. Parcellated sulcal pattern CS3a in the left hemisphere was longer in patients (53.8 ± 25.7 mm vs. 32.7 ± 19.4 mm in controls, *p* < 0.05), while in CS3c it was reversed—longer in controls (52.5 ± 22.5 mm as opposed to 36.2 ± 12.9 mm, n.s. in patients). Non parcellated PCS in the right hemisphere were longer in patients compared to controls (19.4 ± 10.2 mm vs. 12.1 ± 12.4 mm, *p* < 0.001). Therefore, concurrent presence of PCS1 and CS1 in the left hemisphere and to some extent in the right hemisphere may be suggestive of a higher probability of schizophrenia.

## 1. Introduction

Schizophrenia is a heterogeneous group of psychotic diseases with the current absence of clear demarcating neurobiological boundaries of its subtypes [1,2]. Subsequent findings of epidemiological, brain-specific molecular processes and genetic findings may play an important role in the pathogenesis of schizophrenia [3,4]. To this end, aberrations in qualitative anatomical landmarks, i.e., in a form of sulcal morphology, might be of particular value since they may bring additional information on developmental perturbations in regions susceptible to the illness pathology since the gyrification/fissurization of the cortex development is being accomplished principally during the second and third trimester of gestation [5,6]. Abnormalities in the anterior cingulated cortex (ACC) region proven by structural magnetic resonance imaging (MRI) accompanied by neuropathological findings and working memory processing deficit [7] may testify to a neurobiological basis for schizophrenia [8]. Brugger and Howes [9] in their meta-analysis from 2017 based on MRI morphometry from 3901 patients with first-episode schizophrenia and 4040 controls discovered greater homogeneity of the ACC volume and significantly lower mean volume, which may signalize schizophrenia. Modified controllability of functional activity in dorsal ACC may also play an important role in the pathophysiology of schizophrenia, consistent with the importance of this region in cognitive and brain state control operations [10]. At the same time, available findings point to the fact that the ACC gray matter reductions precede psychosis onset and that neurodevelopment of the ACC region might thus be in play [8].

All of the above underlines the importance of quantification of the ACC folding pattern in schizophrenia. Assessment of sulcal patterning in the ACC may serve as a macroscopic probe for hidden developmental events within this area.

Such insights might be expected to arise, especially from the study of the two major sulcal landmarks in the ACC region: the cingulate sulcus (CS) and the paracingulate sulcus (PCS). The CS runs along the corpus callosum and extends posteriorly into the parietal lobe as the marginal ramus. From the dorsal to the anterior portion of the CS, a sulcus that is often present and runs parallel to the CS is referred to as the paracingulate sulcus [11]. The PCS shows significant individual and hemispheric differences. While the CS is present in the majority of cases, the PCS is frequently missing or rudimentary [12]; although it was recently reported to be present in 70.1% of the human population [13]. Leftward asymmetry of PCS has been reported in healthy controls [12,14,15].

Inter-individual variation in the PCS presence is transposed into the prenatal formation of single or double parallel type of the ACC, which is reported in 76% and 24% of adults, respectively [16].

From the developmental perspective, the first appearance of the CS was reported in the 19th week of the fetal period, and the marginal branch of the CS at the 30th ± 3 weeks [17]. An ultrasound study in 677 term-born neonates clearly identified four distinct major folding patterns in the CS [18], which is consistent with reports of the adult brain folding patterns on the MRI and autopsy materials [16,19]. Subsequently, an individual ACC sulcal pattern remains fixed from childhood to adulthood, at the same time that quantitative structural ACC metrics are undergoing profound developmental change [20].

Empirical evidence for the neurodevelopmental and/or neurodegeneration origin of schizophrenia is still a matter of debate [21,22]. Both the CS and the PCS are positioned on the medial sides of brain hemispheres; however, the presence of the CS is constant, unlike the PCS. The CS is located above the callosal body with its rostral end either fusing or tapering off with superior rostral sulcus. Dorsally the CS continues along the callosal body and either abruptly terminates at the level of the central sulcus or continues upward as the marginal ramus towards the hemispheric perimeter. The course of the CS may be uninterrupted or split into one, two, or three distinct segments. The PCS is not present constantly and it is located cranially at the level of the anterior CS; its detailed morphology is irregular [23] (Figure 1).

The upper left picture shows an example of the CS1 type (without interruption), the upper middle shows the CS2 type (with one interruption), and the upper right shows the CS3 type (with two interruptions). The lower left picture shows an example of the PCS0 type (absent), the lower middle shows the PCS1 type (present), and the lower right shows the PCS2 type (prominent). The terms ‘absent’, ‘present’, and ‘prominent’ are according to Yücel’s nomenclature.

Several studies suggest that patients with schizophrenia have a less developed PCS in the left hemisphere compared to controls and may be less likely to have a PCS in either hemisphere [24,25,26,27]. A leftward bias in the PCS asymmetry has been suggested in healthy volunteer studies, and this asymmetry appears to be absent in patients, usually with a reduction in the left PCS prominence [26,28,29]. A well-formed PCS was also less common in high-risk participants compared to controls; however, this association was only present in males [30].

Bilateral absence of the PCS was associated with reductions in reality monitoring performance in healthy individuals with no neurological damage [31]. In one study, hallucinations were associated with specific brain morphology differences in the PCS; a 1 cm reduction in sulcal length in a medial prefrontal cortical fold increased the likelihood of hallucinations by 20%, regardless of the sensory modality in which hallucinations were experienced [32].

Studying the ACC sulcal morphology offers a powerful opportunity to use adult folding patterns to retrospectively assess the differences in earlier brain development between schizophrenia and the norm. Here, we investigate the CS/PCS morphology in both hemispheres in two age/sex matched groups: first-episode schizophrenia and healthy controls. The ACC folding patterns were measured using structural MRI scans.

In our study, we focused on various morphological subtypes of cingulate sulcus parcellations in vivo on the MRI that were not previously investigated in the context of schizophrenia, especially the CS1, CS2a,b, and CS3a,b,c subtypes.

## 2. Materials and Methods

### 2.1. Patient Characteristics and Demographics

We selected 93 patients with first-episode schizophrenia (FES, patients) and 42 age/sex matched healthy participants (controls). Demographic data (Table 1) and schizophrenia patient characteristics and diagnoses (Table 2) are presented below.

The patient sample originated from an Early-Stage Schizophrenia Outcome study (ESO), a prospective trial on FES subjects, conducted at the National Institute of Mental Health, Klecany, Czech Republic (NIMH). FES patients were recruited between 2015 and 2019 through the ESO Patient Enrolment Network, which involves five large psychiatric hospitals in the country (in total 3700 beds), with a total catchment area of 6.5 million people. The inclusion criteria for FES were: (1) the diagnosis of schizophrenia or the diagnosis of an acute polymorphic psychotic disorder, as made by a psychiatrist according to the International Classification of Diseases-10 criteria; (2) the first episode of psychotic illness; and (3) duration of untreated psychosis of less than 24 months. Any patients with psychotic mood disorders (including schizoaffective disorder, bipolar disorder, and unipolar depression with psychotic symptoms) were excluded from the study. All patients were treated with antipsychotic drugs at the time of the MRI scanning. The ESO was approved by the Ethical Committee of the NIMH. The study was carried out in accordance with the latest version of the Declaration of Helsinki.

### 2.2. MR Imaging

All subjects underwent an MR examination on a 3T MRI TRIO scanner (Siemens Medical Systems, Erlangen, Germany) equipped with a 12-channel volume head coil. The protocol of MR imaging included T1-weighted sagittal images obtained using a three-dimensional (3D) magnetization-prepared rapid gradient-echo (MPRAGE) sequence (echo time (TE)/repetition time (TR)/number of acquisitions (NA) = 4.63 ms/2300 ms/1, iPAT = 2, resolution 1 mm × 1 mm × 1 mm); T2-weighted transversal images obtained using 2D Periodically Rotated Overlapping Parallel Lines with Enhanced Reconstruction (BLADE) sequence (TE/TR/NA = 128 ms/5850 ms/1, 2 concatenations, iPAT = 1, resolution 0.6 mm × 0.6 mm × 3 mm); and T2-weighted sagittal images obtained using 3D Fluid Attenuated Inversion Recovery (FLAIR) sequence (TE/TR/NA = 318 ms/6200 ms/1, iPAT = 2, resolution 1 mm × 1 mm × 1 mm). MR images were visually assessed by neuroradiologists to preclude the presence of pathological changes.

### 2.3. Processing of the MRI Images

We accessed the NIMH MRI Hydra database by remote access to the mainframe computer with a standard PC. Within the Hydra database, we worked in Windows X-terminal with Image J software suite (https://imagej.net, accessed on 16 October 2022). MRI images were examined in the sagittal plane and the regions of interest were manually delineated in both hemispheres in the Image J program.

CS and PCS were examined in more than one sagittal section of both brain hemispheres. Due to possible axial movement in the MRI scan and because of variations in the sulcal depth, it was necessary to verify several adjacent sagittal sections to ensure visibility of the entire course of sulci. Figure 1 shows pictures representing sulcal variants purposely depicted in different sagittal sections. Relevant brain structures are presented in more detail in Figure 2.

### 2.4. CS and PCS Neuroanatomy

Generally, the CS is always present in the brain while the PCS may not be present at all. The CS is located on the medial part of both the left and right hemispheres. In its major course, it separates the cingulate gyrus from the gyri located above. It often starts in the frontal lobe at the level of the anterior cingulate gyrus and continues dorsally upward between the paracentral lobule and the praecuneus. The PCS, if present, is located above the CS, usually at the level of the ventral portion of the anterior cingulate gyrus [12,32].

#### 2.4.1. CS and PCS Manual Delineation and Morphology Considerations

Manual delineation of any brain structure requires advanced morphological and topographical expertise for precise delineation of the structures of interest.

The CS starts in the subcallosal area and has an arched course, running above the cingulate gyrus. This way it separates the medial frontal and the parietal cortices from the limbic structure—the cingulate gyrus. Gross delineation of the cingulate sulcus on the MRI is associated with the cingulate gyrus, especially its anterior and posterior portions. The posterior cingulate gyrus is bordered by the marginal ramus of the cingulate sulcus (above), the callosal body (caudally), the parieto-occipital sulcus (dorsally), and the Brodmann area 24 (ventrally). This corresponds to the transition between the anterior and posterior cingulate gyri.

Near the connection between the isthmus of the cingulate gyrus and splenium of the corpus callosum, the marginal branch of the cingulate sulcus reaches the superior end of the hemisphere [33]. According to Terminologia Neuroanatomica [34], we did delineation of the ventral and dorsal borders of the CS in the Cartesian coordinate plane with the center at the level of Monro’s interventricular foramen. We characterized borders in accordance with the following anatomical regions: the ventral anterior cingulate cortex (vACC), the dorsal anterior cingulate cortex (dACC), and the posterior cingulate cortex (PCC). The CS arises as a profound groove in the vACC, from under the genu of callosal body and the rostral gyrus and then it runs parallel to the upper frontal gyrus (Figure 1). The trajectory of the CS is interrupted in the dACC by the dorsal cingulate-frontal infolding. In the PCC, after a horizontal trajectory parallel to the body of the callosal body, CS is interrupted again by the anterior cingulate-parietal connective infolding [35].

#### 2.4.2. Description of CS and PCS Parcellation and Its Nomenclature

First, we measured non-parcellated CS and PCS (without separation of the sulci course into interrupted segments). Next, we performed the segmentation of the CS and PCS.

We measured the linear length of the CS and observed how many parts it was composed of in the case of interruption—in other words, CS morphology was assessed depending on the integrity of the sulcal line. If there was no interruption of the sulcus, we labeled it as CS1, one interruption of the sulcus was labeled as CS2, and two interruptions were labeled as CS3. For the length measurement, two sulcal parts (segments) of CS2 were labeled as CS2a and CS2b, and three sulcal parts of CS3 were labeled as CS3a, CS3b, and CS3c, and their length was measured separately (Table 3). Indexing parts (segments) followed the ventro-dorsal position so that, for example, CS2a was located more ventrally compared to CS2b. The neuroanatomical delineation of the PCS was adapted according to J. R. Garrison’s Paracingulate Sulcus Measurement Protocol [36], based on the original study [32]. We selected the PCS sulcal pattern parcellation identical with Yücel’s nomenclature (three sulcal patterns type—absent, present, and prominent) [14]. PCS morphology was assessed as absent if the length was less than 2 cm (PCS0), present if it extended more than 2 cm (PCS1), and prominent if it was longer than 4 cm (PCS2) (Figure 1).

### 2.5. Statistics

For the purpose of general morphology, we fused all the subparts of each measurement into one number and did the overall statistics. For the more detailed view of morphology, we performed separate statistics for each part of the parcellation of the sulci.

#### 2.5.1. *t*-Test between Groups

Overall length differences between patients with schizophrenia and controls in the non-parcellated PCS and CS were calculated by *t*-test for groups, and separately for cingulate sulcus right (CS R) and left (CS L) and paracingulate sulcus right (PCS R) and left (PCS L).

#### 2.5.2. Mann–Whitney U Test and Kruskal–Wallis Test

The differences in the length of the CS (1, 2, and 3) as well as PCS (0, 1, and 2) between patients with schizophrenia and controls were calculated by nonparametric Mann–Whitney U test and Kruskal–Wallis test (because of unequal numbers of subjects in each category).

#### 2.5.3. ROC Curve Analysis

We used an online ROC analysis web-based calculator with the data format 5 selection. ROC curve analysis was performed for the CS L and the PCS R only (because of the significant *t*-test) (Figure 3).

#### 2.5.4. Chi-Squared Test and Cochran–Mantel–Haenszel Test

The differences between patients with schizophrenia and controls in their distribution within CS and PCS morphology were evaluated separately for the left and right hemispheres by a chi-squared test of the null hypothesis. The Cochran–Mantel–Haenszel test (bilateral) was used to calculate the null hypothesis, similarly to the chi-squared test, but with the sum of the left and right hemispheres results.

#### 2.5.5. Generalized Additive Model

The differences between groups were also assessed in a formalized statistical model of GAM (generalized additive model) [37,38] of semiparametric nature, which allowed for careful adjustment of the group (control versus schizophrenia) effect to sex and age (age effect was modeled by penalized spline with penalty coefficient estimation via generalized cross validation) [39].

GAM and ANOVA/ANCOVA modeling, chi-squared and Cochran–Mantel–Haenszel testing was done in the R environment (R Core Team, Vienna, Austria [40]). Other statistics were calculated in Statistica v.6 software (StatSoft, Tulsa, OK, USA) and the ROC curve analysis by an online program (http://www.rad.jhmi.edu/jeng/javarad/roc/JROCFITi.html, accessed on 16 October 2022). Statistical significance is commented at the 5% level (*p* < 0.05).

#### 2.5.6. Adjustment for Sex and Age

Adjustment for age and sex was performed by the generalized additive model for both patients with schizophrenia and controls for the left and the right side in the CS (1, 2, and 3) and the PCS (0, 1, and 2).

## 3. Results

### 3.1. Length of CS and PCS without Parcellation

Overall length differences of the CS and the PCS are shown in Table 4. The right hemisphere PCS was significantly longer in patients with schizophrenia compared to controls (19.4 ± 10.2 mm vs. 12.1 ± 12.4 mm).

### 3.2. Length of Parcellated CS (1, 2, and 3) and PCS (0, 1, and 2)

Detailed analysis of the morphological subtypes of the CS and the PCS and their lengths in schizophrenia patients and controls are shown in Table 3. The length of CS3a in the left hemisphere was significantly longer in patients with schizophrenia (53.8 ± 25.7 vs. 32.7 ± 19.4 mm in controls) but in CS3c it was reversed—longer in controls (52.5 ± 22.5 mm) compared to schizophrenia patients (36.2 ± 12.9 mm), but without significance. The length of PCS0 in the right hemisphere was significantly longer in patients with schizophrenia (11.7 ± 5.2 mm) compared to controls (6.2 ± 6 mm) (it was not present at all).

### 3.3. Incidence and Distribution of Patients and Controls between CS and PCS Morphology Types

Table 5 shows the differences in numbers of patients with schizophrenia and controls for each CS and PCS morphology type and incidence of distribution of particular sulcal patterns within CS and PCS morphology. Higher incidences of appearance in patients compared to controls in CS1, PCS0, and PCS1 were observed in both hemispheres; a lower incidence in patients was observed in CS3 in both hemispheres; in CS2 on the left there was more incidence in controls compared to patients, and the opposite held true on the right—more in patients compared to controls (PCS2 in both hemispheres was not present).

The number of patients with schizophrenia who had concurrently present CS1 as well as PCS1 in both hemispheres was 9 (8.7%). The number of patients with PCS1 L and concurrently present CS1 L was 25 (83%); PCS1 R and concurrently, CS1 R was 28 (68%).

### 3.4. Adjustment for Age and Sex Performed by Generalized Additive Model

After the adjustment for age and sex, we found PCS on the right side to be significantly (*p* < 0.001) larger for schizophrenic patients than for controls (by about 7.346 on average). PCS on the left was not significantly different between patients and controls (*p* = 0.304). Similarly, CS did not differ significantly between patients and controls neither on the right (*p* = 0.375), nor on the left side (*p* = 0.877). The difference between PCS on the right and left (laterality of the PCS) was significantly larger for patients than for controls (*p* = 0.019)—by about 5.587 on average. Analogous laterality for CS was not significantly different between patients and healthy controls for CS (*p* = 0.331).

### 3.5. ROC Analysis Evaluation

ROC analysis was performed for statistically significant results of the length of the CS L and PCS R between patients and controls. For the CS L were estimates of binomial ROC parameters A = 0.097 with standard error (SE) (A) = 0.24, B = 1.36 with SE (B) = 0.19, and correlation (A, B) = 0.032. The area under the fitted curve was 0.52 with SE = 0.056, trapezoidal (Wilcoxon) area = 0.51 with estimated SE = 0.054. For the PCS, R were estimates of binomial ROC parameters A = 0.63 with SE (A) = 0.31, B = 1.58 with SE (B) = 0.26, and correlation (A, B) = 0.14. The area under the fitted curve was 0.63 with SE = 0.06, trapezoidal (Wilcoxon) area = 0.63 with estimated SE = 0.055 (Figure 3). Therefore, ROC analysis suggests that the practical classification schizophrenia patients and controls based only on the length of the CS L and PCS R is (obviously) far from being perfect.

## 4. Discussion

Numerous studies in the literature document the relationship between the morphological changes of the central nervous system and schizophrenia, e.g., cortical areas [41], amygdala [42], hippocampus [43], and others [44], but they are not suitable for morphological diagnosis of schizophrenia because of their non-specificity and frequent anatomical variability.

The CS and the PCS are two regions found within the ACC on the medial side of the brain hemisphere. Morphological variability of the PCS was described, for example, in [32]. Specificity of the morphological changes in ACC vs. other cortical areas were described by Fornito et al. [8]. Variability in the CS organization in human adult and fetal cadavers was described by Marinescu et al. [35]. The CS structure was analyzed in six groups of adult brains whose medial hemisphere was parcellated into I-III sectors, clockwise to the CS course. The major differences in the CS organization could be summarized into the following observations: in sector I the subgenual part of callosal body varies in the numbers and shape of the CS infoldings; in sector II there is broken trajectory of the CS by the dorsal cingulate-frontal infolding; and in sector III the trajectory is sinusoid and there is a convoluted aspect of the marginal branch. It is still difficult to account for these variations on the MRI where tiny details of the CS neuroanatomy are often blurred by the arachnoid.

### 4.1. Distribution of CS and PCS Morphological Patterns

We found significant differences between schizophrenia patients and controls in all types of the CS (1, 2, and 3) and the PCS (0, 1, and 2). Left-right asymmetries were not significant with the exception of PCS0 and PCS1 in schizophrenia patients. The most common in schizophrenia patients and controls was the presence of PCS0 morphology, followed by PCS1 and PCS2. Schizophrenia patients were found to have the most common CS1 morphology, followed by CS2 and CS3. This was in contrast with the control group, where the most common morphology was CS3 and CS2, followed by CS1. At least for the distribution of the morphological patterns of both PCS and CS, there are significant differences between schizophrenia patients and controls. The trend of these morphological distributions in healthy subjects described by Wei et al. [12] was similar to our results (although the effects of sex and handedness were included as well).

### 4.2. Length of the PCS

We found significantly longer non-parcellated PCS in the right hemisphere of schizophrenia patients compared to the control group. This is based on statistically significant difference only in the right PCS0 part in schizophrenia (see Table 3). Schizophrenia patients were reported to have a frequent absence of the left PCS. In healthy volunteers, prominent or present PCS was described more frequently in the left hemisphere compared to the right. On the contrary, schizophrenia patients were found to have no significant asymmetry and types prominent or absent. These all were found with the same frequency [27,28]. In our study, we found more often similar PCS1 in controls in the left hemisphere compared to the right. However, we did not observe enough cases of PCS2 (prominent) to create valid statements about their frequency neither in control nor in schizophrenia patients. We observed *left-right asymmetry in the PCS length in schizophrenia patients only* and not in the control group. Leftward asymmetry was observed only in PCS0 (absent) and rightward asymmetry in PCS1 (present). This is also in contrast to the study by Le Prevost et al. [28]. Artiges et al. [45] measured the morphology of the PCS in relation to fMRI hypoactivation in the ACC in 13 patients with schizophrenia and 16 healthy controls. They did *not observe a difference between healthy subjects and patients with the PCS* (PCS1 and PCS2 in our classification), but patients with schizophrenia exhibited significant hypoactivation of the ACC, where the PCS was absent (PCS0 in our classification). According to our data, *PCS0 were present in both schizophrenia patients and controls* with the highest statistical occurrence, so that we support the notion that hypoactivation may take place even in the size of our studied group (patients with schizophrenia *n* = 93 and control group *n* = 42). The advantage of the Image J program is manual delineation, by which it is possible to distinguish the length of the PCS0 from 0 to 2 cm.

Shorter PCS (without parcellation) was observed in patients diagnosed with psychotic disease having hallucinations; they had shorter PCS compared to non-hallucinating psychotics and healthy controls [46].

In summary, we show that there is a *reverse tendency in the PCS0 and PCS1 length between schizophrenia patients and controls*. In PCS0 type morphology (“absent” type according to Yücel’s nomenclature), we found a lower number of schizophrenic patients with the absent PCS compared to controls, regardless of the hemisphere (left 67% patients and 81% controls; right 55% patients and 76% controls). This finding is consistent with reports of [27,28]. We found the opposite for PCS1—a higher number of patients with schizophrenia having PCS1 type of morphology (“present” type according to Yücel’s nomenclature) (left 32% patients and 19% controls; right 44% patients and 19% controls) compared to controls. These differences may explain the inconsistency between the observed leftward PCS hemispheric asymmetries in schizophrenia and healthy controls [47,48] versus reduced PCS asymmetries in schizophrenia patients [27,28] because of the intrinsic differences in both the CS and the PCS.

### 4.3. Length of the CS

It appears in schizophrenic patients that the CS is generally uninterrupted when viewed from the sagittal section, while in the control group this was rarely seen. On the contrary, the control group was almost equally split into half of participants having one and two interruptions of the CS. In other words, in the CS there is a reverse trend in the number of participants as for increase/decrease of the number of interruptions (in control groups the number of interruptions rises, while in schizophrenia patients it declines). It appears that it is an advantage to have an interrupted CS as opposed to the uninterrupted CS.

In summary, we show that there is a difference in length between CS1 and CS2 in schizophrenia patients and controls. In CS1 type morphology (“uninterrupted” type), we found a higher number of patients with schizophrenia with uninterrupted CS compared to controls, regardless of the hemisphere (left—66% patients and 14% controls; right—62% patients and 12% controls). On the contrary, in CS2 (one interruption) there was a higher number of controls compared to patients (left—41% controls and 26% patients; right—43% controls and 32% patients). In CS3, similarly to CS2, there was a higher number of controls than patients (left—45% controls and 6% patients; right—45% controls and 5% patients).

The length of the *left CS3a was significantly longer in patients with schizophrenia*, while the *length of CS3c was longer in controls*; CS3b did not differ between the groups. From the perspective of the study by Marinescu et al. [35], CS3a corresponds to sector I and CS3c to sector III. CS3a is close to the rostral area of the callosal body with the variability of connective infolding between the gyrus cinguli and the parolfactory area, and the infolding between gyrus cinguli and the rostral gyrus that continues with the gyrus frontalis superior. CS3c is the continuation of the main sulcal line into the marginal part of CS that also shows variability in the posterior cingulate-parietal connective infolding [35].

The length of the CS and PCS in the healthy population was described, for example, by Fornito et al. [47]; however, there are no such data in schizophrenia patients. We did not observe any gross morphological abnormalities of the CS and the PCS in schizophrenia patients, so we presume that the overall shape and morphology are not valid as clinical markers for the differential diagnosis of schizophrenia. On the contrary, the CS and the PCS length differ between both groups significantly, specifically, a decrease in length of the CS in schizophrenia patients is compensated by the length increase in the PCS. We found this valid for the right hemisphere only. Interestingly, we discovered two opposite trends in the left hemisphere. The length of CS3a was significantly longer in schizophrenia patients, while CS3c was significantly longer in controls. Our results partially correspond to the hypothesis of Rametti et al. [23], who studied the maximal depth and volume of the anterior cingulate sulcus and the PCS in 23 patients and 24 controls. However, they do not define the borders of the main measured area (the ACC) precisely in their study. The other difference is they used automated voxel-based morphometry measurements with manual correction, while we did all the measurements manually. The study found significant decrease in volume in the anterior cingulate sulcus in the left hemisphere and significant increase in the PCS volume in the right hemisphere of the patients with schizophrenia compared to controls [23].

The limitation of our study is the analysis of the sulcal shape that was performed in the sagittal plane view only. This way we may have missed invaginations of the CS in the white matter in a lateral direction. This could have led to the labeling of some sulci (CS2 and CS3) as interrupted; although they could have been present, if viewed on the coronal sections. Furthermore, general atrophy of the white matter in schizophrenia patients could lead to a merging of sulcus as a result of smoothening and unification of the surface cortical areas [49]. This observation could be supported by controls, who have relatively longer, although interrupted, courses of the CS. Another limitation is the demographic differences between the control group and the schizophrenia group; thus, analyzing a larger population sample may improve the understanding of CS and PCS morphology.

Delineation of the CS and PCS was done on MRI scans in sagittal projection—this is what clinicians would use, too. The problem with delineation of the ACC is that it is based on the cytoarchitectonic, post-mortem diagnosis. The posterior part of the ACC in the transition to the posterior cingulate gyrus (PCC) is difficult to evaluate on the MRI [50]. For this reason, it is not usable as a clinical diagnostic tool. We could not account for precise anatomical details of both sulci delineation, as described by, for example, Marinescu et al. [35], because their study was done on post-mortem brain tissue with many more anatomical details. Nevertheless, we described the Methods section in great detail so that it can be used as guidance for both sulci delineation on the MRI scans, which would minimize bias in interpretation of their occasionally variable course.

We analyzed each response separately, without corrections to multiple comparisons. This corresponds to univariate/marginal analyses and might lead to some inflation of falsely positive results. As we intend the study to be exploratory, or as a pilot study in the uncovered area, we do not want to be overly conservative. On the other hand, it is certainly in place to verify the significant findings on independent samples in the future.

Regarding the difference in educational attainment between the schizophrenia group and controls, other studies have shown that patients with schizophrenia have lower educational attainment than those without [51,52]. Attempts to create a pool of schizophrenia patients with the same level of education as controls is referred to in the literature as the “matching fallacy”, as it can lead to the selection of atypical groups, such as patients with high education or, conversely, controls with low education [53,54].

### 4.4. Practical Use of CS and PCS Length as a Support Tool for Schizophrenia Diagnosis

The data in Table 3 could be useful as a guide for further MRI studies of schizophrenia. Depending on the MRI morphological typology of the CS and the PCS subsets in patients, it is possible to predict the following combinations of the medial hemisphere sulcal appearance: (1) absent PCS and one of the CS type (1–3); (2) present PCS and one of the CS type (1–3); (3) prominent PCS and one of the CS type (1–3). Out of these, only the concurrent unilateral presence of PCS1 and CS1 in the left or the right hemisphere suggests a higher probability of schizophrenia. All other combinations point at the lack of disease. The morphological differences that we documented may potentially be of use as support tools demonstrating the differences present in schizophrenia. However, further research is required prior to implementing morphological measures as a support tool for schizophrenia diagnosis.

Brain ultrasound and magnetic resonance images of fetuses at 18–23 gestational weeks showing the structure of the sulcus cinguli suggest the possibility of comparing morphology between adult brains and possible changes in the ACC region from the prenatal period [55]. The deviation in cortical folding within the cingulate area suggests neurodevelopmental alterations within this area, albeit incomplete understanding of cellular, genetic, and experience-dependent plasticity behind aberrant cortical folding precludes, for the time being, conclusive statements about the neurobiological underpinnings of those changes. Given the current status of knowledge, further elucidation of the role of genes involved in driving the maturational trajectories of cortical patterning and the impact of events that disrupt fetal neurodevelopment are prerequisite for the whole understanding of those processes inflicting upon the composition of cortical architecture in schizophrenia.

## 5. Conclusions

Our study expands previously documented morphological classification of the sulcal patterns within the anterior cingulate area, especially the quantification of various morphology types of cingulate and paracingulate sulcus in patients with schizophrenia and the healthy population. Based on our results on ACC parcellation (CS1, 2, and 3; PCS0, 1, and 2), it is possible to postulate that the concurrent presence of PCS1 and CS1 in the left hemisphere as well as, to some extent, in the right hemisphere suggests a higher probability of schizophrenia. However, concurrent presence of PCS1 and CS1 in the left and right hemisphere at once is not very frequent.

## Figures and Tables

**Figure 1 jcm-12-00033-f001:**
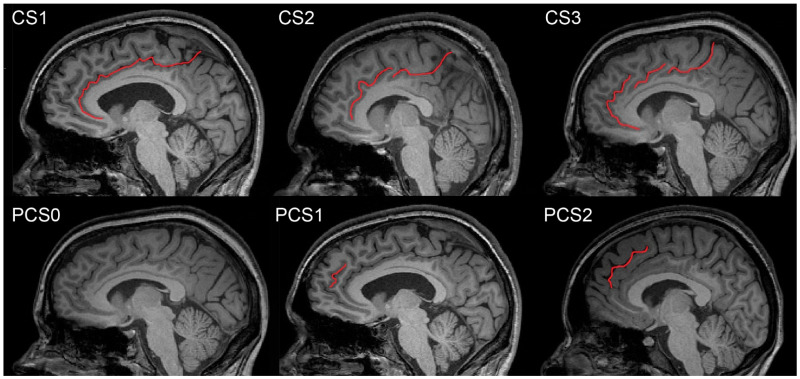
Illustrative examples of possible appearances of CS and PCS morphology types on the MRI images.

**Figure 2 jcm-12-00033-f002:**
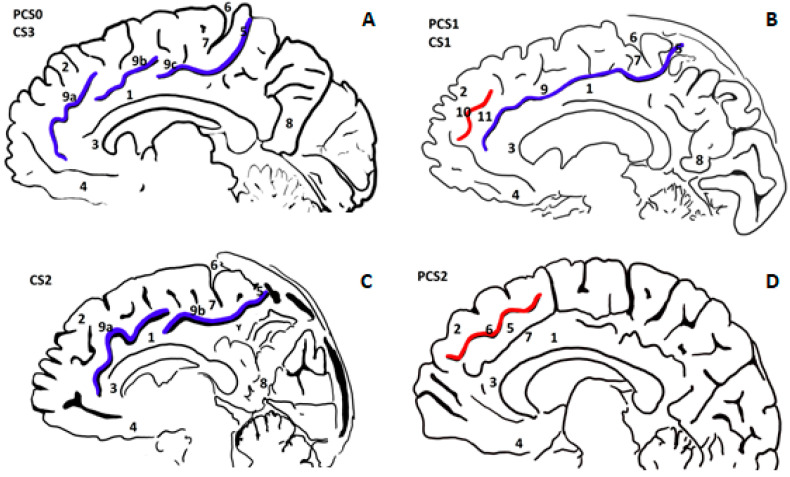
CS and PCS delineations on the medial hemisphere (right) of the brain. Combinations PCS0 and CS3, PCS1 and CS1, PCS2 alone, and CS2 alone do not reflect their co-occurrence in the patients; they were clustered together for better illustration. PCS0 means there is no paracingulate sulcus or its length is shorter than 2 cm. (**A**) PCS0 and CS3: 1—cingulate sulcus, 2—superior frontal gyrus, 3—genu corporis callosi, 4—gyrus rectus, 5—the marginal branch of cingulate sulcus, 6—central sulcus, 7—paracentral lobule, 8—precuneus gyrus, 9a—cingulate sulcus (part one, CS3a), 9b—cingulate sulcus (part two, CS3b) and 9c—cingulate sulcus (part three, CS3c); (**B**) PCS1 and CS1: 1—cingulate sulcus, 2—superior frontal gyrus, 3—genu corporis callosi, 4—gyrus rectus, 5—the marginal branch of cingulate sulcus, 6—central sulcus, 7—paracentral lobule, 8—precuneus gyrus, 9—cingulate sulcus (without interruption, CS1), 10—paracingulate sulcus, 11—paracingulate gyrus (without interruption, PCS1); (**C**) CS2: 1—cingulate sulcus, 2—superior frontal gyrus, 3—genu corporis callosi, 4—gyrus rectus, 5—the marginal branch of cingulate sulcus, 6—central sulcus, 7—paracentral lobule, 8—precuneus gyrus, 9a—cingulate sulcus (part one, CS2a), 9b—cingulate sulcus (part two, CS2b); (**D**) PCS2: 1—cingulate sulcus, 2—superior frontal gyrus, 3—genu corporis callosi, 4—gyrus rectus, 5—paracingulate gyrus, 6—paracingulate sulcus, 7—cingulate sulcus.

**Figure 3 jcm-12-00033-f003:**
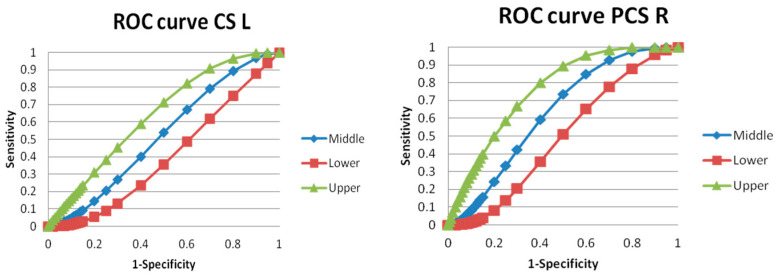
ROC curve analysis for length differences in left cingulate sulcus (CS L) and right paracingulate sulcus (PCS R) in schizophrenia patients vs. controls without morphological parcellation. The area under the fitted curve was 0.52 with SE = 0.056 (CS) and 0.63 with SE = 0.06 (PCS).

**Table 1 jcm-12-00033-t001:** Demographic Data of Schizophrenia Patients and Healthy Controls. The Data are Presented as Mean Values ± SD.

Demographic Data	Patients Sample Total (*n* = 93)	Healthy Controls (*n* = 42)
Sex, *n* = male/female	47/46	17/25
Age at baseline MRI (V1), year	30.1 ± 7.4	31.6 ± 6.2
Age at second, follow up MRI (V2), year	31.3 ± 7.4	32.6 ± 5.6
Years of education at V2	14.8 ± 3.1 *	17.9 ± 3.6 *
Interscan interval (V1–V2), month	13.0 ± 1.8	16.0 ± 3.8

Values between groups were not significant, except for years of education (* *p* < 0.0001).

**Table 2 jcm-12-00033-t002:** Characteristics of the Patients and Diagnoses. The Data are Presented as Mean Values ± SD.

Detailed Overview of Patient’s Examination	
Age at onset, year	29.4 ± 7.3
Duration of illness (DUI) at baseline V1, month	7.7 ± 11.2
Duration of untreated psychosis (DUP), month	4 ± 11.9
Global Assessment of Functioning (GAF) at V1	65.6 ± 15.4
Global Assessment of Functioning (GAF) at V2	79.9 ± 13.0
Positive and Negative Syndrome Scale (PANSS) negative 1	15.7 ± 5.8
Positive and Negative Syndrome Scale (PANSS) negative 2	13.3 ± 5.4
Positive and Negative Syndrome Scale (PANSS) 1Σ	60 ± 16.0
Positive and Negative Syndrome Scale (PANSS) 2Σ	48.2 ± 14.6
**Final Diagnosis (M/F)**	
Schizophrenia	29/30
Acute polymorphic psychotic disorder without symptoms of schizophrenia	3/2
Acute polymorphic psychotic disorder with symptoms of schizophrenia	9/11
Acute schizophrenia-like psychotic disorder	2/0
Other acute predominantly delusional psychotic disorders	1/0
Schizoaffective disorder	1/5

**Table 3 jcm-12-00033-t003:** Detailed analysis of the length of the parcellated CS and PCS on the left and right side in schizophrenia patients and controls and their incidences. The data are presented for all types of morphology (CS types 1, 2, 3 and PCS types 0, 1, 2). The length data are presented as mean values in mm ± SD. The incidence is presented as a number of cases (*n*) and their percentages. Incidences of CS types 2 and 3 are given as a single value for all relevant sulcal parts (segments). Statistical significance was calculated by nonparametric Mann–Whitney U test and Kruskal–Wallis test (*p* < 0.05 and *p* < 0.001).

Brain Structures	Patients		Controls	
	Length in mm	Incidence *n* (%)	Length in mm	Incidence *n* (%)
Left cingulate sulcus type 1 (CS1)	110.7 ± 15.8	61 (66)	105.2 ± 21.7	6 (14)
Left cingulate sulcus type 2a (CS2a)	59.8 ± 16.3	26 (28)	52 ± 19.3	17 (41)
Left cingulate sulcus type 2b (CS2b)	55.9 ± 18.3		62.9 ± 26	
Left cingulate sulcus type 3a (CS3a)	53.8 ± 25.7 *p* < 0.05	6 (6)	32.7 ± 19.4 *p* < 0.05	19 (45)
Left cingulate sulcus type 3b (CS3b)	20.5 ± 9		22.3 ± 12.9	
Left cingulate sulcus type 3c (CS3c)	36.2 ± 12.9		52.5 ± 22.5	
Right cingulate sulcus type 1 (CS1)	113.3 ± 18.7	58 (63)	123.2 ± 19.1	5 (12)
Right cingulate sulcus type 2a (CS2a)	65.9 ± 23.8	30 (32)	57.2 ± 26.7	18 (43)
Right cingulate sulcus type 2b (CS2b)	53.6 ± 2.3		63.8 ± 27.5	
Right cingulate sulcus type 3a (CS3a)	47.7 ± 7.7	5 (5)	35.9 ± 16.1	19 (45)
Right cingulate sulcus type 3b (CS3b)	39.2 ± 17.2		24.9 ± 19.7	
Right cingulate sulcus type 3c (CS3c)	34.5 ± 16.3		49.3 ± 20.1	
Left paracingulate sulcus type 0 (PCS0)	10.8 ± 4.8	63 (68)	10.9 ± 4.3	34 (81)
Left paracingulate sulcus type 1 (PCS1)	27.1 ± 4.7	30 (32)	29.3 ± 6.5	8 (19)
Left paracingulate sulcus type 2 (PCS2)	N/A	0 (0)	N/A	0 (0)
Right paracingulate sulcus type 0 (PCS0)	11.7 ± 5.2 *p* < 0.001	51 (55)	6.2 ± 6 *p* < 0.001	32 (76)
Right paracingulate sulcus type 1 (PSC1)	28.4 ± 5.2	41 (44)	28.2 ± 6.3	8 (19)
Right paracingulate sulcus type 2 (PCS2)	47.7 ± N/A	1 (1)	41.6 ± 2	2 (5)

**Table 4 jcm-12-00033-t004:** Left and right CS and PCS without parcellation overall length differences between schizophrenia patients and control groups. The data are presented as mean values ± SD in mm. Statistical significance was calculated by *t*-test for groups.

Brain Structures	Patients Length (mm)	Controls Length (mm)	*p*-Values
Left cingulate sulcus	112.4 ± 16.3	111.5 ± 18.8	n.s.
Right cingulate sulcus	114.7 ± 22.7	116.6 ± 24.4	n.s.
Left paracingulate sulcus	16.1 ± 9.1	14.4 ± 8.7	n.s.
Right paracingulate sulcus	19.4 ± 10.2	12.1 ± 12.4	*p* < 0.001

n.s. = non significant.

**Table 5 jcm-12-00033-t005:** Comparison of patients and controls with CS1, 2, and 3 and PCS0, 1, and 2 types of morphology based on Table 3. The numbers show incidences (absolute values) of a particular morphology type in between patients with schizophrenia and controls. N/A—not applicable due to zero or low numbers of presence.

Group.	CS1 Left/Right	CS2 Left/Right	CS3 Left/Right	PCS0 Left/Right	PCS1 Left/Right	PCS2 Left/Right
Patients	61/58	2/30	6/5	63/51	30/41	N/A
Controls	6/5	17/19	19/19	34/32	8/8	N/A

## Data Availability

Not applicable.

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
