# Peer review of "Morphology of Anterior Cingulate Cortex and Its Relation to Schizophrenia"

_jcm, 2022, doi:10.3390/jcm12010033_

Round 1

Reviewer 1 Report (New Reviewer)

This is a good paper describing the potential correlation between anterior cingulate changes and schizophrenia. The work is well written even if I think there could be some improvements for the authors:

- the abstract should be unstructured;

- There are too many spaces in the text with line wraps which make it more difficult to read. Please reduce the number of spaces

- The tab. 2 is not clearly visible as a grid is missing. Please add it;

- please consider the following references:

1) Li Q, Yao L, You W, Liu J, Deng S, Li B, Luo L, Zhao Y, Wang Y, Wang Y, Zhang Q, Long F, Sweeney JA, Gu S, Li F, Gong Q. Controllability of Functional Brain Networks and Its Clinical Significance in First-Episode Schizophrenia. Schizophr Bull. 2022 Nov 19:sbac177. doi: 10.1093/schbul/sbac177. Epub ahead of print. PMID: 36402458.

2)Feng N, Palaniyappan L, Robbins TW, Cao L, Fang S, Luo X, Wang X, Luo Q. Working memory processing deficit associated with a nonlinear response pattern of the anterior cingulate cortex in first-episode and drug-naïve schizophrenia. Neuropsychopharmacology. 2022 Nov 14. doi: 10.1038/s41386-022-01499-8. Epub ahead of print. PMID: 36376466.

3) Aquila I, Sacco MA, Ricci C, Gratteri S, Ricci P. Quarantine of the Covid-19 pandemic in suicide: A psychological autopsy. Med Leg J. 2020 Dec;88(4):182-184. doi: 10.1177/0025817220923691. Epub 2020 May 15. PMID: 32410517.

Author Response

- the abstract should be unstructured;

Corrected

- There are too many spaces in the text with line wraps which make it more difficult to read. Please reduce the number of spaces

We hope we corrected all line wraps as suggested.

- The tab. 2 is not clearly visible as a grid is missing. Please add it;

Added

- please consider the following references:

References Li et al., 2022 as well as Feng et al., 2022 were added to the manuscript; reference Aquila et al., 2020 is not related to the topic of the article so it was not added.

Reviewer 2 Report (New Reviewer)

This study focused on the morphology of anterior cingulate cortex and its relations with schizophrenia. This topic is meaningful and interesting. My main concerns are as below:

1. What are the advantages of Image J software compared with the more widely used SPM, FSL, or FreeSurfer, which is able to perform ROI analyses in the ACC as well?

2. Would it be possible to provide some potential correlations between neuroimaging changes and symptom severity (e. g. PANSS) and other clinical variables?

3. The “Discussion” chapter could also highlight your results compared with previous findings using other software.

Author Response

Ad 1) What are the advantages of Image J software compared with the more widely used SPM, FSL, or FreeSurfer, which is able to perform ROI analyses in the ACC as well?

The advantage of Image J is that it offers manual delineation of the ROI; for example, FreeSurfer is automated software that compares shades of grey/white with possible discrepancies, especially in the cortical areas of the brain. We tried FreeSurfer but it was not possible to recognize enough structure of various sulci in the grey matter of the brain. For example, a big problem is the nonexistent paracingulate sulcus (PCS0 in our article) which is not recognized by the machine, but also split cingulate sulcus is not correctly recognized. Similarly in the article from 2010: Destrieux C, Fischl B, Dale A, Halgren E. Automatic parcellation of human cortical gyri and sulci using standard anatomical nomenclature. Neuroimage. 2010 Oct 15;53(1):1-15. doi: 10.1016/j.neuroimage.2010.06.010.

The Image J program is an application that is widely diffused in the whole community due to its versatility and effectiveness. For last two years it was cited in 1059 papers, while FiJi (more accurate version of Image J was used in 631 papers).

Ad2) Would it be possible to provide some potential correlations between neuroimaging changes and symptom severity (e. g. PANSS) and other clinical variables?

The study was narrowly focused on neuroanatomical abnormalities in schizophrenia compared to healthy controls. However, we consider the inclusion of clinical characteristics to be highly relevant and beneficial. Concerning PANSS score we were unable to estimate for the controls. So it was not possible to use it in this model, because one variable was missing.

Cingulate cortex structure and function might be related to symptomatology in schizophrenia, namely the magnitude of negative symptoms. Those findings altogether may point to abnormal neurodevelopment of this brain area.

BERSANI, Francesco Saverio, et al. Cingulate Cortex in Schizophrenia: its relation with negative symptoms and psychotic onset. A review study. Eur Rev Med Pharmacol Sci, 2014, 18.22: 3354-3367.

HAATVEIT, Beathe, et al. Divergent relationship between brain structure and cognitive functioning in patients with prominent negative symptomatology. Psychiatry Research: Neuroimaging, 2021, 307: 111233.

LI, Huabing, et al. Reduced connectivity in anterior cingulate cortex as an early predictor for treatment response in drug-naive, first-episode schizophrenia: a global-brain functional connectivity analysis. Schizophrenia research, 2020, 215: 337-343.

Ad3) The “Discussion” chapter could also highlight your results compared with previous findings using other software.

All relevant comparisons of other studies software or methodologies with our study were italicized in the Discussion section and also minor addendum was done in the Discussion section.

Reviewer 3 Report (New Reviewer)

Author Response

General concerns:

  1. The findings are based upon a comparison between 93 patients with schizophrenia or schizophrenia spectrum patients and 42 healthy controls.  However, the healthy controls differ demographically from the experimental group in two important ways: (1) the controls are better educated (potentially different in cognitive abilities?), and (2) there was a much greater percentage of females in the control group.  It is hard to understand why the controls were not age, sex, and educational achievement matched with the schizophrenic patients. The first MRI study of neuroanatomy had the same flaw and that resulted in some retractions of findings because they did not hold up when an appropriate control group was used.

As for differences between sex, there is no significant difference between the gender distribution (p-value=0.3693). But as for the education there is a highly significant difference between the length of education for controls and patients (p-value<0.0001). Schizophrenia appears as early as 19-20 years of age, so it definitely prevents many patients from achieving higher education. Therefore, comparing by years of education would mean that we would have a small number of schizophrenia patients who are outside the median (norm). So the median level of education for controls must necessarily be higher than for schizophrenia, as the table shows. We added to the Table 1 and also to the Legend p value for the diference in years of education.

  1. There is inadequate description in the Introduction and Discussion of the importance of studying anatomical variations in the cingulate sulcus or the paracingulate sulcus. The significance of this manuscript is totally dependent on knowing that these anatomical variations have a relevant functional consequence.

We added specific sentence to the Introduction section „Modified controllability of functional activity in dorsal ACC may also play important role in the pathophysiology of schizophrenia, consistent with the importance of this region in cognitive and brain state control operations“ with citation Li, Q., Yao, L., You, W., Liu, J., Deng, S., Li, B., Luo, L., Zhao, Y., Wang, Y., Wang, Y., Zhang, Q., Long, F., Sweeney, J.A., Gu, S., Li, F., and Gong, Q. (2022). Controllability of Functional Brain Networks and Its Clinical Significance in First-Episode Schizophrenia. Schizophr. Bull. Advance online publication. doi: 10.1093/schbul/sbac177. Also we added explanatory sentence about potential importance of sulcal morphology in schizophrenia in the Discussion section “Brain ultrasound and magnetic resonance images of fetuses 18-23 gestational weeks showing the structure of the sulcus cinguli suggest the possibility of comparing morphology between adult brains and possible changes in the ACC region from the prenatal period” with citation Ghai, S., Fong, K.W., Toi, A., Chitayat, D., Pantazi, S., Blaser, S. (2006). Prenatal US and MR imaging findings of lissencephaly: review of fetal cerebral sulcal development. Radiographics. 26, 389-405. doi: 10.1148/rg.262055059

  1. Speculation on the use of these findings to assist in the diagnosis of schizophrenia are premature. The manuscript does not present any data on the specificity or sensitivity of the findings toward making a diagnosis of schizophrenia.

In our manuscript, we merely point out the differences between the healthy controls and schizophrenia patients. As schizophrenia diagnosis is made clinically, potential morphological correlates may be of use in the future; however, further research is certainly required. We clarified that in our latest manuscript version.

  1. Is it possible that the findings reflect differences based on neurocognitive differences and not schizophrenia? It would be helpful to analyze if educational achievement determines the findings.

Statistical analysis shows that the length of study (education) did not affect the findings.

Specific concerns:

Ad 1. In Table 2, it mentions that the duration of untreated psychosis was only 4 months on average. It is not defined whether that only applies to the time before antipsychotic medications were initiated or also includes any time the patient was untreated because of non-compliance.

Duration of untreated psychosis (DUP) information was collected during the first study visit (V1), and was defined as the time elapsed between psychosis onset and antipsychotic treatment initiation, more in Morgan et al., 2006. (Morgan C, Abdul-Al R, Lappin JM, Jones P, Fearon P, Leese M, et al. Clinical and social determinants of duration of untreated psychosis in the AESOP first-episode psychosis study. Br J Psychiatry 2006; 189: 446–52.)

We defined the onset of psychosis as one week or more of one of the following symptoms: delusions, hallucinations, grossly disorganized or catatonic behavior, or disorganized speech, based on the framework proposed by Craig et al. 2000. (Craig TJ, Bromet EJ, Fennig S, Tanenberg-Karant M, Lavelle J, Galambos N. Is there an association between duration of untreated psychosis and 24-month clinical outcome in a first-admission series? Am J Psychiatry 2000; 157: 60–6.)

For our study, we have estimated the onset of psychosis based on information from medical or hospitalization reports from treating psychiatrists—if available—, as well as from the report of the clinician who addressed the patient to the study. If the psychosis time of onset could not be determined based on these reports, further clinical assessments were carried out to estimate the date of onset of sustained positive psychotic symptoms based on the information obtained in the patient interview. Where appropriate, this date has been updated over the course of the ESO prospective study if new information became available which allowed more precise identification of the onset date. A final DUP value was made on the basis of all available prospective data gathered as part of the clinical file after entry into the ESO study.

Ad 2. It would be helpful to include a reference that explains the difference between schizophrenia and an “acute polymorphic psychotic disorder”.

Acute Polymorphic Psychotic Disorder (ICD-10) represents a psychotic disorder with an acute onset (within two weeks), presenting thought and perception disorders variable in a range of hours. The polymorphism and instability are characteristic of the overall clinical picture, and the duration of psychotic features does not justify a diagnosis of schizophrenia. ICD-10 offers two subcategories of APPD. If APPD  is accompanied by symptoms typical of schizophrenia for the majority of the time, „APPD with symptoms of schizophrenia“ is recognized. Otherwise, APPD without schizophrenia symptoms is diagnosed.  The duration of APPD with schizophrenic symptoms is limited to 1 month, and the longer duration of first-rank symptoms of schizophrenia warrants rediagnosing to schizophrenia.

Ad 3. It does not appear that the statistical analyses corrected for multiple comparisons.

We analyzed each response separately, without corrections to multiple comparisons. This corresponds to univariate/marginal analyses and might lead to some inflation of falsely positive results. As we intent the study as exploratory, or pilot study in the uncovered area, we do not want to be overly conservative. On the other hand, it is certainly in place to verify the significant findings on independent sample in future.

Ad 4. In the Discussion, the limitation of significant differences in the control group demographics should be mentioned.

As per this suggestion, a sentence regarding the demographic differences has been added to the limitations of the study in the Discussion section of the manuscript.

Round 2

Reviewer 3 Report (New Reviewer)

I have two remaining concerns:

1.     The explanation offered about why it is appropriate to have differences in the educational level of the schizophrenic patients and controls is not valid in my opinion.  The fact that schizophrenic patients do not achieve as expected does not excuse ignoring the inherent bias that occurs when the controls have a higher achievement level.  The manuscript at the minimum should include a discussion of this potential flaw in the design.

2.     The manuscript needs to specifically state that statistical analyses were not corrected for multiple comparisons.  Also, the results of correction for multiple comparisons should be described so that the reader has the opportunity to determine the influence of the stated findings have validity.

Author Response

We thank reviewer for stimulating and challenging comments.

For the point 1)

We have added a paragraph to the Discussion section explaining the lack of "multiple comparisons" statistics.

For the point 2)

We have added a paragraph to the Discussion section explaining the difference between the level of education in schizophrenia patients and controls.

Please see below:

Matching for education level in studies of schizophrenia.

Patients with schizophrenia achieve lower levels of education than people without (Tesli et al., 2022; Crossley et al., 2022).

Tesli M, Degerud E, Plana-Ripoll O, Gustavson K, Torvik FA, Ystrom E, Ask H, Tesli N, Høye A, Stoltenberg C, Reichborn-Kjennerud T, Nesvåg R, Naess Ø. Educational attainment and mortality in schizophrenia. Acta Psychiatr Scand. 2022 May;145(5):481-493. doi: 10.1111/acps.13407.

Crossley NA, Alliende LM, Czepielewski LS, Aceituno D, Castañeda CP, Diaz C, Iruretagoyena B, Mena C, Mena C, Ramirez-Mahaluf JP, Tepper A, Vasquez J, Fonseca L, Machado V, Hernández CE, Vargas-Upegui C, Gomez-Cruz G, Kobayashi-Romero LF, Moncada-Habib T, Arango C, Barch DM, Carter C, Correll CU, Freimer NB, McGuire P, Evans-Lacko S, Undurraga E, Bressan R, Gama CS, Lopez-Jaramillo C, de la Fuente-Sandoval C, Gonzalez-Valderrama A, Undurraga J, Gadelha A. The enduring gap in educational attainment in schizophrenia according to the past 50 years of published research: a systematic review and meta-analysis. Lancet Psychiatry. 2022 Jul;9(7):565-573. doi: 10.1016/S2215-0366(22)00121-3. 

The attempt to create groups with the same level of education between a set of schizophrenic patients and controls is discussed in many papers and referred to as the "matching fallacy". The debate on this topic has been ongoing since 1992, where an article (Resnik 1992) points out, based on a study (Goldberg et al., 1990), that deliberately selecting schizophrenia patients having a level of education comparable to controls may result in selection of atypical groups of high-achieving patients or low-achieving controls. Furthermore, attempts to statistically correct for education may be misleading because educational attainment will be confounded with age of onset, duration, and severity of illness. A similar argument is made in a study by (Kremen et al., 1995).

Resnick SM. Matching for Education in Studies of Schizophrenia. Arch Gen Psychiatry. 1992;49(3):246. doi:10.1001/archpsyc.1992.01820030078011

Kremen WS, Seidman LJ, Faraone SV, Pepple JR, Lyons MJ, Tsuang MT. The '3 Rs' and neuropsychological function in schizophrenia: a test of the matching fallacy in biological relatives. Psychiatry Res. 1995 Mar 27;56(2):135-43. doi: 10.1016/0165-1781(94)02652-1

This manuscript is a resubmission of an earlier submission. The following is a list of the peer review reports and author responses from that submission.

Round 1

Reviewer 1 Report

Reviewer 1 Comments: 

The authors investigate the cortical folding of anterior singular cortex regions with the first episode of schizophrenic and control subjects using MRI. The authors measured the length of parcellated and non-parcellated CS, PCS. They found that the parcellated sulci pattern in the left hemisphere (CS3a) is significant in patients compared to control subjects, but in CS3c, it was reversed in controls. The authors conclude that the presence of PCS1 and CS1 in the left hemisphere and, to some extent, in the right hemisphere may point to some probability of schizophrenia in patients. While the authors provide interesting and potentially valuable findings, there are a few concerns as described below. 

1.         In Figure 1, the representative images are from the same subjects or a single subject? The upper and lower images look similar to the reviewer’s eyes.

2.         The reviewer is wondering how the authors arrived at the correct middle sagittal slice section of MRI images. Is there any landmark to get exact sagittal images in all the subjects, even a small oblique orientation or angle might change the sulci pattern in the sagittal images? 

3.         Is there any correlation between age and sex vs. CS and PCS morphology?

4.         Since the authors have collected both T1 and T2 sagittal images, did the authors find similar sulci morphology in the T2 weighted images? Did the authors compare with T2 weighted images when they measured CS and PCS length? 

5.         Due to the geometrical complexity and the variability of cortical folds across subjects, manual tracing of the human brain will be challenging. So the sensitivity of the manual trace of the human brain can lead to human error or bias. Thus, the author believes that results from manual tracing are less accurate and repeatable. So it is hard to convince an interesting reader that the findings are strong. The authors have already collected MP-RAGE T1 measurements for all the subjects. It is not that painful to do additional processing using automated volumetric-based morphometry (VBM) measurements to measure the brain morphology in those regions. 

6.         It would be nice to have gray- and white-matter volumes and total brain tissue volume for both subject groups. It has been reported that brain folding is related to its volume. 

7.         The difference in length between CS and PCS in some brain structures is so small, and manual delineation may lead to some bias or human error. Please clarify. 

8.         The authors did not explain how the parcellated and non-parcellated regions were delineated in the results section. Can the authors elaborate more on that?

9.         In Table 4, why did the right paracingulate sulcus show significance while other regions did not. Please provide a rational explanation in the discussion section.

It would be more convincing and strengthen the author’s findings if the above concerns were addressed.

Reviewer 2 Report

I commend the authors for conducting a detail study of the ACC patterns in patient with first episodes schizophrenia.

I have described my questions and concerns topic-wise in order of the manuscript:

Abstract:

-Sentence 18 of method section is not structured correctly.

-Please describe the type of research study you performed.

-Similar to the conclusion in the manuscript, the described results do not provide appropriate or sufficient conclusive remarks for eg. What do these differences in CS and PCS sulcal patterns suggest clinically, or biologically, or for future research. The conclusion in the abstract should give readers a quick and better idea of what one can take away from the study. Expansion of prior results is elusive.

Manuscript:

Introduction:

-Plagiarism- Statements 31,32,33, 38,39,49,41-43 have phrases/sentences exactly similar to those in the abstracts for the quoted citations (3, 7, 8) suggesting plagiarism.

-Previous studies have been conducted to study differences in ACC patterns in schizophrenia and first episode psychosis patients. How is your study different? How does it add to the current knowledge? Please mention this explicitly.

Methods:

-It would be helpful for readers to have a diagram for the descriptions in section 2.4.1 for CS and PCS and surrounding areas.

Analysis:

1). Please adjust for cortical volume or surface area in the differences of lengths of CS and PCS as differences in cortical volume or surface areas can impact the length of the sulci.

2). Why were the clinical scores for psychotic symptoms not compared to the CS/PCS sulcal morphology/length distribution and differences.

3). There were two MRIs performed. Please clarify from which MRIs were the sulcal scores measured. Please also describe the reason for choosing initial vs follow up MRI scores.

4)Sentences 164 and 165 need a change in structure to convey the correct meaning.

5)Though the difference in the length of CS3a on left in patient compared to control was significant, the difference in CS3c was nonsignificant. Please make sure to mention results that are nonsignificant if you end up describing the difference in incidence. Otherwise, readers will have a false impression. Would suggest to make similar statements for section 3.3 as these were also nonsignificant. For the same reason, please convey the differences in Table 5 were nonsignificant as you did for table 1. What does N/A stand for in Table 5. Please make sure to add legends for your table for easy readability.:

Discussion:

What is the biological or clinical interpretation of your results?